# Exploring the Palynological, Chemical, and Bioactive Properties of Non-Studied Bee Pollen and Honey from Morocco

**DOI:** 10.3390/molecules27185777

**Published:** 2022-09-07

**Authors:** Meryem Bakour, Hassan Laaroussi, Pedro Ferreira-Santos, Zlatina Genisheva, Driss Ousaaid, José Antonio Teixeira, Badiaa Lyoussi

**Affiliations:** 1Laboratory of Natural Substances, Pharmacology, Environment, Modeling, Health, and Quality of Life (SNAMOPEQ), Department of Biology, Faculty of Sciences Dhar Mehraz, Sidi Mohamed Ben Abdellah University, Fez 30000, Morocco; 2CEB—Centre of Biological Engineering, University of Minho, Campus de Gualtar, 4710-057 Braga, Portugal; 3LABBELS—Associate Laboratory, 4710-057 Braga, Portugal

**Keywords:** bee products, physicochemical characterization, antioxidant activity, polyphenols, antihyperglycemic activity, nutritional values

## Abstract

Bee products are known for their beneficial properties widely used in complementary medicine. This study aims to unveil the physicochemical, nutritional value, and phenolic profile of bee pollen and honey collected from Boulemane–Morocco, and to evaluate their antioxidant and antihyperglycemic activity. The results indicate that *Citrus aurantium* pollen grains were the majority pollen in both samples. Bee pollen was richer in proteins than honey while the inverse was observed for carbohydrate content. Potassium and calcium were the predominant minerals in the studied samples. Seven similar phenolic compounds were found in honey and bee pollen. Three phenolic compounds were identified only in honey (catechin, caffeic acid, vanillic acid) and six phenolic compounds were identified only in bee pollen (hesperidin, cinnamic acid, apigenin, rutin, chlorogenic acid, kaempferol). Naringin is the predominant phenolic in honey while hesperidin is predominant in bee pollen. The results of bioactivities revealed that bee pollen exhibited stronger antioxidant activity and effective α-amylase and α-glycosidase inhibitory action. These bee products show interesting nutritional and bioactive capabilities due to their chemical constituents. These features may allow these bee products to be used in food formulation, as functional and bioactive ingredients, as well as the potential for the nutraceutical sector.

## 1. Introduction

Honey is the most popular bee product that is easily accessible and commercially available. It is known for a wide range of pharmacological properties including antimicrobial, antioxidant, anti-inflammatory, hypoglycemic, and cardio-protective, among others [1]. The main composition of honey is carbohydrates (60–85%) and water (12–23%). It contains also other functional compounds, for instance, minerals, vitamins, amino acids, organic acids, polyphenols, flavonoids, and enzymes, as well as pollen grains [2]. Similarly, bee pollen is an important bee product that is commercially available. This hive product is formed by the agglutination of pollen grains with the bee’s salivary secretions, nectar and/or honey, and enzymes [3]. The bee pollen contains an average of 25.7% reducing sugars, 22.7% of proteins, 30.8% of digestible carbohydrates, 5.1% of lipids, and phenolic compounds representing an average of 1.6% [4]. Bee pollen is known for its health promising effects and displayed several pharmacological properties such as antioxidant, anti-inflammatory, and antiproliferative effects [5,6,7].

To verify the authenticity and to detect impaired honey and bee pollen, many screening tools are used for routine quality control such as physicochemical characteristics, organoleptic characteristics, sugar composition, proline content, and hydroxymethylfurfural (HMF) content [8,9,10]. Additionally, the determination of the nutritional value, the identification, and the quantification of bioactive markers such as phenolic compounds as well as the determination of the antioxidant capacity are necessary parameters for the standardization of these bee products and the evaluation of their safety to use as food supplements.

Citrus species are one of the most melliferous plants attractive to bees due to their high pollen production [11]. Bee pollen and citrus honey are bee-hive products largely produced and consumed in the Mediterranean areas including north African and European countries [12]. This monofloral honey has a particular color, taste, aroma, and flavor, which is associated with specific chemical composition. In addition to organoleptic and sensorial characteristics, monofloral honey has specific physical and chemical properties. Preliminary screen analyzes of organic honey samples with a predominance of citrus pollen from Morocco confirmed acceptable microbial and good physicochemical quality [12,13,14,15,16].

As far as we know, this is the first study that evaluates these honey and bee pollen samples collected from the same apiaries installed in the Boulemane area, Morocco. In this context, the present work aimed to study the nutritional values, physicochemical characteristics, biocompound composition (phenolics, proteins, minerals, and sugar contents), structural characterization (ATR-FTIR), antioxidant activity, and the in vitro inhibition of α-amylase and α-glucosidase of these two bee products (honey and bee pollen).

## 2. Materials and Methods

### 2.1. Bee Pollen and Honey Samples

Bee pollen and honey were harvested in July 2019 from a sedentary apiary composed of twenty-nine hives. The sampling was carried out as follows: 29 honey samples were harvested from each hive to obtain 250 g and 29 bee pollen samples were collected from each hive to obtain an amount of 290 g. Honey and bee pollen samples were harvested and kept in food-grade jars and stored until analysis (3 months) at 4 °C for honey and in a cool, dry, dark place for bee pollen. The information about the apiary and its geographical location was presented in Table 1.

### 2.2. Melissopalynological Analysis

The pollen grains spectrum analysis in bee pollen and honey was determined as described elsewhere [17,18]. Depending on the percentage of pollen grains in each sample, the following classification was used: predominant pollen (if represents more than 45% of the pollen grains counted), secondary pollen (if represents an amount between 16% and 45% of the pollen grains counted), important minor pollen (if represents an amount between 3% and 15% of the pollen grains counted), and minor pollen (if represents an amount less than 3% of the pollen grains counted) [18].

### 2.3. Physicochemical Parameters of Honey and Bee Pollen

#### 2.3.1. Electrical Conductivity

Twenty grams of thyme honey was dissolved in 100 mL of distilled water, the conductivity was measured at 20 °C using an electrical conductivity cell (model 4510, Jenway, UK). The result was expressed as µS/cm [19].

#### 2.3.2. PH

The pH was measured in a solution of honey or bee pollen (10 g of honey or bee pollen dissolved in 100 mL of ultra-pure water) using a pH meter (OHAUS ST2100-F, Parsippany, NJ, USA) [5,19].

#### 2.3.3. Free Acidity, Lactone Acidity, and Total Acidity

Free acidity was measured using 1 g of honey dissolved in 25 mL of ultra-pure water, then a solution of NaOH (0.05 M) was added until the equivalence point pHe = 8.3, while the value of lactone acidity was obtained after the addition of 1 mL of NaOH 0.05 M followed by the titration with HCl (0.05 M) to return to the equivalence point. The total acidity value is the sum of free acidity and lactone acidity [19].

#### 2.3.4. Ash Content

Five grams of honey were placed in a furnace at 600 °C until constant mass, and then the weight of ash was measured [19].

#### 2.3.5. Moisture and Total Soluble Solids (TSS)

Moisture and total soluble solids (TSS) were determined using a refractometer (PCE-5890, PCE Instruments, Southampton, UK) according to the standard method AOAC (n°52.729) [20].

#### 2.3.6. Diastase Activity

Diastase activity was analyzed as follows: a mixture of 2 mL of honey solution (1 g of honey was dissolved in 1.5 mL of distilled water mixed with 500 μL of acetate buffer (pH 5.3), and 300 μL of sodium chloride) and 2 mL of starch solution (2 g of starch were dissolved in 90 mL of distilled water, boiled for 3 min, and then terminated in the 100 mL line with distilled water) were separately put in the bath at 40 °C for 15 min, and then 1 mL of the starch solution was added to the 2 mL of honey extract, and the timer was started. After 2 min, 100 μL of the mixture were taken and added to 1 mL of the iodine solution (4 g of iodine potassium were added to 400 μL of iodine stock and filled up to 100 mL with distilled water) and 4 mL of distilled water. The absorbance of the reaction mixture was read at 660 nm. The negative control was prepared by using a mixture of starch solution and distilled water. Diastase activity was calculated using the formula (1):(1)$$\mathrm{Diastase}\;\mathrm{Number}=\frac{300}{Tx}$$

*Tx*: the time it took the reaction for the absorbance of the blue color to decrease to approximately 0.235. The results were expressed as Schade units/gram of honey [19].

#### 2.3.7. Honey Color

The color of honey was estimated by determining the absorbance at 635 nm using a UV/VIS spectrophotometer (Synergy HT, BioTek Instruments, Inc., Winooski, VT, USA). For that, 10 g of the honey was dissolved in 20 mL of distilled water. The *mm Pfund* values were obtained using the following formula (2) [21]:(2)mmPfund=−38.7+371.39×Absorbance

#### 2.3.8. Melanoidins Content

Melanoidin content was determined based on the browning index by measuring the net absorbance of the honey at 450 nm and 720 nm (net absorbance = A_450_–A_720_) using a UV/VIS spectrophotometer (Synergy HT, BioTek Instruments, Inc., Winooski, VT, USA) [22]. The melanoidin content was in absorption units.

#### 2.3.9. Water Activity

The water activity (A_w_) of bee pollen was measured using a water activity meter (model ms1, Novasina AG, Lachen, Switzerland) [23].

#### 2.3.10. Total Protein (TP)

The protein content of bee pollen or honey was estimated by quantification of total nitrogen after sample acid (HNO_3_) digestion using a Kjeldahl digestor (Tecator, FOSS, Hillerød, Denmark), applying the nitrogen conversion factor (N × 6.25) [24]. The results were expressed as a gram of total protein per 100 g of bee pollen or honey.

#### 2.3.11. Mineral Content

Mineral elements were obtained by the calcination method using inductively coupled plasma atomic emission spectroscopy (ICP-AES) (Activa Horiba Jobin Yvon-Ovou 1048, France). Mineral elements were determined using an air/acetylene flame, the quantitative determination was carried out after calibrating the instrument using ranges of calibrations of K, Ca, Na, Mg, Fe, Cu, Zn, Pb, Cd dissolved in 0.1% lanthanum. Honey and bee pollen samples were analyzed in triplicate [25].

### 2.4. Structural Characterization by ATR-FTIR Spectroscopy

Functional groups and bonding arrangement of constituents present in the raw bee pollen and honey were determined by Fourier transform infrared spectroscopy (FTIR) using an ALPHA II- Bruker spectrometer (Ettlingen, Germany) with a diamond-composite attenuated total reflectance (ATR) cell. The FTIR spectra were recorded in the range of 4000–400 cm^−1^, with 60 scan cycles per sample at a resolution of 4 cm^−1^ [26].

### 2.5. Biocompounds Determination of Bee Pollen and Honey

#### 2.5.1. Total Phenolic Content (TPC) and Total Flavonoid Content (TFC)

The content of total phenolic and total flavonoid in honey and bee pollen was determined as described in our previous studies [26,27].

The concentration of TPC was measured using the Folin–Ciocalteu method, which is based on the colorimetric reduction/oxidation reaction of phenols. So, 5 µL of sample or water for control were mixed with Folin–Ciocalteu reagent (15 µL) and 60 µL of sodium carbonate (75 g/L). The reaction was performed for 5 min at 60 °C, and absorbance was measured at 700 nm by a UV/VIS spectrophotometer (Synergy HT, BioTek Instruments, Inc., USA). Gallic acid was used to perform a calibration curve (R^2^ = 0.994), and TPC values were expressed as gallic acid equivalents (GAE) (mg GAE/g).

For the determination of TFC, a spectrophotometric assay based on the formation of an aluminum chloride complex was used. Thus, 500 µL of the sample or water for control was mixed with distilled water and 5% sodium nitrite solution. After, AlCl_3_ 10% solution was added, and thereafter NaOH 4% solution was added to the mixture. Then, the mixture was properly mixed and allowed to stand for 15 min, and the absorbance was measured at 510 nm. Quercetin was used to perform the standard curve (R^2^ = 0.995) and the TFC results were expressed as mg of quercetin equivalents (QE) per g of honey or bee pollen (mg QE/g).

#### 2.5.2. Identification and Quantification of Individual Phenolic Compounds by UHPLC

Honey and ethanolic extract of bee pollen were analyzed using a Shimadzu Nexpera X2 UPLC chromatograph equipped with Diode Array Detector (DAD) (Shimadzu, SPD-M20A). Separation was performed on a reversed-phase Aquity UPLC BEH C18 column (2.1 mm × 100 mm, 1.7 μm particle size; from Waters) and a precolumn of the same material at 40 °C. The flow rate was 0.4 mL/min. HPLC grade solvents water/formic acid 0.1% and acetonitrile were used [26]. Phenolic compounds were identified by comparing their UV spectra and retention times with that of corresponding standards.

Catechin (≥98% of purity), caffeic acid (≥98%), vanillic acid (≥97.0%), *o*-coumaric acid (≥97.0%), ferulic acid (≥99.0%), ellagic acid (≥95.0%), naringin (≥95.0%), hesperidin (≥97.0%), cinnamic acid (≥99.0%), resveratrol (≥99.0%), rosmarinic acid (≥98.0%), quercetin (≥95.0%), apigenin (≥99.0%), rutin (≥94.0%), chlorogenic acid (≥95.0%), and kaempferol (≥99.0%). All used standards were of analytical grade (purity level above 94%) and procured from Sigma Aldrich (St. Louis, MO, USA).

Quantification was carried out using calibration curves for each compound analyzed using concentrations between 250–2.5 mg/L. In all cases, the coefficient of linear correlation was R^2^ > 0.99. Compounds were quantified and identified at different wavelengths (209–370 nm). The values of individual phenolic compounds were expressed in milligrams per kilogram of samples (mg/kg). All analyses were made in triplicate.

#### 2.5.3. Soluble Proteins (SP)

Soluble proteins were determined using the Bradford method [28]. Bovine serum albumin (BSA, 2000–50 µg/mL) was used to perform the calibration curve (R^2^ = 0.985). The soluble protein content was expressed as a gram of BSA equivalents (BSAE) per 100 g of bee pollen or honey (g BSAE/100 g).

#### 2.5.4. Total Carbohydrates (TC)

Total carbohydrates were analyzed as follows: a mixture of 50 μL of bee pollen or honey, 150 μL of sulfuric acid (96–98% (*v*/*v*)), and 30 μL of phenol reagent (5%) was heated for 5 min at 90 °C. After cooling down at room temperature for 5 min, the absorbance of the mixture was measured at 490 nm by a microplate reader. Glucose (10–600 mg/L) was used as a standard to achieve the calibration curve (R^2^ = 0.995). The total carbohydrate content was expressed as a gram of glucose equivalents (GlcE) per 100 g of bee pollen or honey (g GLcE/100 g) [24].

#### 2.5.5. Quantification of Individual Sugars, Furfural, and Hydroxymethylfurfural (HMF)

The concentrations of glucose, fructose, sucrose, furfural, and HMF in the bee products samples were determined by HPLC using a BioRad Aminex HPX-87H column (300 × 7.8 mm) with a gel particle size of 9 μm, eluted at 60 °C with 0.005 M sulfuric acid and a flow rate of 0.6 mL/min. The peaks corresponding to sugars were detected using the Knauer IR intelligent refractive index detector, whereas HMF and furfural were detected using a Knauer UV detector set at 210 nm [29]. Quantification was carried out using calibration curves for each compound (R^2^ > 0.99). All analyses were made in triplicate.

### 2.6. Antioxidant Activity

Three different methods of measuring the antioxidant activity were used: DPPH, ABTS, and FRAP.

Free radical scavenging activity by the DPPH method was determined as follows: 270 μL of 2,2-diphenyl-1-picryl-hydrazyl-hydrate (DPPH) solution (150 μM, prepared in methanol with an absorbance of 0.700 ± 0.01 at 515 nm) was mixed with 30 µL of different concentrations of honey (200 to 1000 μg/mL) or bee pollen extract (16 to 260 μg/mL). Then the absorbance of the mixture reactions was measured at 515 nm after 1 h of incubation in the dark [26]. The antiradical activity (% inhibition) was calculated using Equation (3). The concentration of bee pollen or honey required to scavenge 50% of DPPH (IC_50_) was determined graphically using the curve plotted by the percentage of DPPH inhibition as a function of the sample concentration. The IC_50_ values were expressed in µg/mL. 6-hydroxy-2,5,7,8-tetramethylchroman-2-carboxylic acid (Trolox) was used as a positive control.
(3)$$\%\;\mathrm{Inhibition}=\frac{\mathrm{Abs}\;\mathrm{control}-\mathrm{Abs}\;\mathrm{sample}}{\mathrm{Abs}\;\mathrm{control}}\times 100$$

Radical cation decolorization (ABTS assay) was determined as follows: 200 μL of 2,2′-azino-bis(3-ethylbenzothiazoline-6-sulfonic acid) diammonium salt (ABTS) radical cation solution was mixed with 10 µL of different concentrations of bee pollen (65 to 1040 μg/mL) and honey (from 1000 to 8000 μg/mL). The mixture was incubated in the dark for 30 min, and then the absorbance was measured at 734 nm. Trolox was used as a positive control. The ABTS radical cation inhibition percent was determined using Equation (3). The IC_50_ results were expressed in µg/mL [26].

Ferric reducing antioxidant power (FRAP assay) was determined as follows: 10 μL of different concentrations of honey (400 to 1800 mg/mL) or bee pollen extract (0.03 to 1.04 mg/mL) was mixed with 290μL of FRAP reagent (pH 3.6). Then, the mixture was incubated at 37 °C for 15 min. The absorbance is determined at 593 nm [26]. An aqueous solution of ferrous sulfate was used to build the calibration curve. FRAP values are expressed as micromoles of ferrous equivalent per g material (μmol Fe^2+^/g sample).

### 2.7. Antihyperglycemic Activity

To assess the antihyperglycemic capacity of honey and bee pollen, two important enzymatic assays were performed: α-Glucosidase and α-amylase inhibitory activities. For α-glucosidase inhibition assay, a mixture of different honey (1 to 6 mg/mL) or bee pollen (0.065 to 2.08 mg/mL) concentrations and p-nitrophenyl-R-d-glucopyranoside (pNPG, 3 mM) was added to the α-glucosidase solution (10 U/mL), and after 15 min of incubation at 37 °C, the reaction was stopped by adding Na_2_CO_3_ solution (1 M). The intensity of p-nitrophenol coloration produced was measured at 400 nm [27].

For α-amylase inhibition assay, a mixture of 500 µL of α-amylase solution (0.5 mg/mL) and 500 µL of different concentrations of honey (1 to 6 mg/mL) or bee pollen (0.065 to 2.08 mg/mL) was incubated at 37 °C for 15 min. Distilled water and ethanol 70% were used as a negative control, and acarbose was used as a positive control. Then, 500 µL of starch solution (1%) was added and the mixture was incubated for 15 min at 37 °C. Immediately, 1 mL of dinitrosalicylic acid color reagent was added to the reaction and placed for 10 min in a boiling water bath. The final mixture was diluted 10 times and the absorbance of each dilution was read at 540 nm [27].

Equation (3) was used to calculate α-amylase and α-glucosidase inhibitory activity (%). The honey or bee pollen concentration required to inhibit 50% (IC_50_) of α-amylase and α-glucosidase activities were calculated from a dose–response curve, and the results were expressed in mg/mL.

### 2.8. Statistical Analysis

The data obtained are presented as mean ± standard deviation (SD) values. GraphPad Prism software (version 6.0; GraphPad Software, Inc., San Diego, CA, USA) was used for statistical analyses. A student *t*-test was used to compare honey and bee pollen samples, *p* < 0.05.

## 3. Results and Discussion

### 3.1. Melissopalynological Analysis

The results of the melissopalynological analysis presented in Table 2 showed that the predominant pollen found in both honey and bee pollen was *Citrus aurantium*, 48%, and 50%, respectively. When the percentage of predominant pollen was over 45%, the samples were classified as monofloral [30]. These results were expected, as the honey and bee pollen samples were collected from the same apiary in Boulemane, Morocco, which is an area rich in citrus plants.

### 3.2. Physicochemical Analysis of Honey and Bee Pollen

Physicochemical parameters are necessary analysis routines to check the good quality, reveal the adulteration of bee products, and know their geographical and botanical origin [8,31,32,33]. In this study, honey and bee pollen collected from the same hives were analyzed for various potential physicochemical parameters (Table 3). The moisture analysis showed a percentage of 3.34 ± 0.02% in bee pollen and 20.08 ± 0.03% in honey. The percentage of moisture in bee pollen revealed that the sample was dried because the fresh bee pollen should contain a percentage of water ranging between 20% and 30%, and the high content of water makes fresh bee pollen an ideal culture medium for microorganisms [34]. For that, it is recommended to dry it until obtaining less than 6% of humidity [35]. On the other hand, the value of moisture obtained in our examined honey exceeded slightly the maximum limit fixed by the Codex Alimentarius Commission at 20% [33]. The obtained result could be due to the precocious collection of honey before its total maturity, which constitutes a favorable environment for mold and yeast development when it largely exceeds 20% [36]. In addition to water content, pH is another parameter that influences indirectly the shelf life and thus the stability of bee pollen and organic honey. In addition to the botanical and pedo-climatic characteristics of each harvest station, this parameter is also affected by conditioning storage and beekeepers’ practices. As documented in Table 3, for pH analysis, no significant differences were shown between honey and bee pollen (4.17 ± 0.04 and 4.35 ± 0.03, respectively). These results are in agreement with those published by Adaškevičiūtė and coworkers [37] for eighteen dried bee pollen samples and eleven honey samples harvested from twelve countries in which pH values ranged between 4.30 and 5.22 in honey, and between 3.72 and 4.74 in bee pollen.

The ash content was higher in bee pollen than in honey, with values of 3.13 ± 0.03% and 0.36 ± 0.01%, respectively. The ash content is an important quality parameter because it reflects the inorganic (minerals) content in these bee products. It may be influenced by the botanical origin and the soil type [38]. The results obtained by our samples fulfill the limits of the standards that set a maximum limit of 4% for bee pollen [17] and 0.6% for honey [39]. Moreover, previous data found by our research group indicate that the ash content of Moroccan citrus honey is below the maximum value specified by the codex Alimentarius and European Unit Council [33,40]. For instance, Aazza and coworkers characterized 17 commercialized honey samples, in which citrus honey had an ash value of 0.15 ± 0.01% [15]. Additionally, El Menyiy et al. [13] investigated the physicochemical analysis of 14 Moroccan monofloral honeys and reported that citrus honey collected from the Sidi kacem station had an ash content of 0.07 ± 0.006% (see Table 4). Regarding bee pollen, it is necessary to mention that, to date, there is only one published work investigating the nutritional quality and the physicochemical characterization of eight monofloral bee pollens collected from different localities of Morocco [5]. Accordingly, ash content varied between 1.81 ± 0.10% and 4.22 ± 0.08%, for *Reseda luteola* (60%) bee pollen and *Coriandrum sativum* (70%), respectively.

Collected bee pollen contains a high amount of energetic ingredients including, carbohydrates and proteins. Nowadays, the search for new cheaper, nutritive, healthier, and sustainable protein sources from non-animal origins is driven by a leading tendency and attract the attention of many researchers and health care companies worldwide. Current data showed that the bee pollen is richer in proteins compared to honey, at 27.53% vs. 0.380%, respectively. The total protein content of investigated bee pollen was in agreement with the international standards which fixed a minimum value of 10 g/100 g and a maximum value of 40 g/100 g of bee pollen dry weight [5,35].

Furfural was not detected in both honey and bee pollen samples. Hydroxymethylfurfural (HMF), or 5-hydroxymethyl 2-furaldehyde, is a water-soluble organic compound derived from sugars. It is produced during the thermal heating of honey as a result of dehydration of fructose and glucose. HMF is generally recognized as a pilot parameter reflecting the heating historic and thus honey freshness [41].

The analysis of HMF in honey revealed content of 18.63 ± 0.22 mg/kg (Table 3), which is in the range of Moroccan citrus honey that shows HMF content between 5.01 and 43.3, between 5.05 and 43.30; and from 0.08 to 32.60 mg/kg for samples harvested from Berkane, Northwest and Nador areas, respectively (Review Table 4) [14,16,42]. While, in bee pollen, HMF was not detected. Moreover, the HMF content must not exceed 40 mg/kg in honey [33], validating the quality of our analyzed product. Bee pollen was subjected to water activity analysis and it showed a value of 0.30 ± 0.01, which is in agreement with the ones reported by Estevinho and coworkers for twenty-two Portuguese organic bee pollen samples (0.21–0.37) [43]. This value indicates that our sample is not subjected to fermentation, since a water activity below 0.60 is considered insufficient for the growth of osmophilic yeasts, which is typical for preventing microbiological spoilage (yeasts, bacteria, and fungi) and reducing deteriorative chemical and biochemical reactions [34,44].

Furthermore, honey was analyzed for other physicochemical parameters necessary to check its good quality, such as a diastase activity that revealed a value of 12.64 ± 1.25 Schade units/g. This result was in line with those reported for Moroccan citrus honey collected from Berkane, North-west, and Nador stations in which values ranged from 1.63 to 29.0, from 1.61 to 287, and from 4.3 to 11.0 Schade units/g, respectively (see Table 4) [14,16,42].

The freshness of honey is determined by the analysis of diastase and HMF. Diastase is an enzyme added by bees in honey to break down starch into glucose, and a low diastase value indicates enzyme degradation due to the overheating of honey. Likewise, a high HMF value indicates that the honey is stored in poor conditions, aged, or overheated [31,38,45]. Additionally, the acidity of honey is a very important quality parameter that is largely influenced by honey age and its specific composition of aromatic and aliphatic acids. High acidity indicates microbial deterioration of honey or a high content of water which causes fermentation of the honey [46,47]. Obtained results showed that the honey’s total acidity was 33.20 ± 1.76 mEq/kg, the free acidity was 24.13 ± 1.75 mEq/kg, and the lactonic acidity was 9.07 ± 1.69 mEq/kg. These values are in line with those recommended by the Codex Alimentarius Commission [33].

The value of TSS obtained in honey was 79.69 ± 0.02%, thus the studied honey can be considered of high grade and highly stable during storage [48]. In addition to that, the analysis of electrical conductivity is a very important criterion to reveal the botanical origin of different kinds of honey and to differentiate between blossom and honeydew honey. It depends on the ash and acid contents of honey [31]. The current result showed a value of 614.66 ± 3.78 µS/cm, which is in the range of electrical conductivity values revealed for the blossom honey [49].

The honey color is the first quality parameter appreciated and evaluated by consumers. Generally, the darkest honey is known for its good quality [50]. The most commonly used technique for color determination is based on the optical comparison, using a Pfund color scale [51]. The Pfund value obtained by analyzing honey showed a value of 142.78 ± 4.26 mm, thus according to the Pfund scale, the honey is classified as dark amber (Pfund > 114 mm) [52].

Melanoidins are high molecular weight compounds produced in the later stages of the Maillard reaction [53]. It was proven that melanoidin formation increased after heat treatment of honey [54]. The analysis of melanoidins in honey showed a content of 0.93 ± 0.01, this result is in line with the range of color standard designation [48].

From a dietary and energy standpoint, carbohydrates and proteins represent the major sources of honey bees’ nutrition and are considered principal constituents of honey and bee pollen. Results presented in Table 5 showed that honey and bee pollen proteins are highly soluble (0.375 ± 0.001 vs. 27.00 ± 4.00 g BSA/100 g, respectively), compared to the results obtained for total protein (Table 3). For total carbohydrates and individual sugars content, the honey was richer than bee pollen (Table 5). The concentration of total carbohydrates was 71.52 ± 0.33 g Glceq/100 g in honey and 31.69 ± 0.95 g Glceq/100 g in bee pollen. The total carbohydrate content found in the investigated honey exceeded the minimum value fixed by both international regulations, codex Alimentarius and EU Council at 65 mg/kg (see Table 4) [33,40]. Fructose was the major individual sugar in both bee products, with a value of 36.76 ± 3.30 g/100 g in honey and 16.42 ± 0.09 g/100 g in bee pollen followed by glucose with a concentration of 26.08 ± 2.71 g/100 g and 11.44 ± 0.12 g/100 g for honey and bee pollen, respectively. In honey, the fructose/glucose ratio (F/G) influences the physical state and the crystallization of honey. In general, F/G is the best-used index for classifying honey crystallization, in which honey is defined as slow or absent when F/G > 1.33, medium 1.11 ≤ F/G ≤ 1.33, and fast when F/G < 1.11 [55]. Honey with an F/G greater than 1.35% is always in a liquid state even when stored for a long time. The F/G of the evaluated honey (1.41%) reaffirms its liquid character. In addition to its impact on the physical state and sensory characteristics of honey, the F/G ratio continues to be a crucial variable that condition or even restricts the use of honey in many critical physiological situations, such as glucose and lipid metabolic dysfunctions. Previous scientific data documented that fructose contained in honey decreased markedly the fasting blood glucose level in diabetic rat models and improve the lipid status of diabetes patients [56,57]. For that, in addition to antioxidant molecules and many other bio-valuable micro/macronutrients, fructose-rich honey might be at least useful to enhance human physiological abilities and prevents several metabolic disorders including dyslipidemia and diabetes.

Regarding sucrose, honey had a concentration of 2.93 ± 0.35 g/100 g vs. 1.38 ± 0.03 g/100 g in bee pollen (Table 5). The sucrose content of the tested honey sample was higher than that found in eighteen Moroccan Zantaz honeys, in which sucrose content was below 0.2 g/100 g [58]. Moreover, Aazza and coworkers [15] examined Seventeen monofloral Moroccan samples and showed values between 0.85 ± 0.06 g/100 g in carob honey and 3.72 ± 0.06 g/100 g in eucalyptus honey (Review Table 4). Although, a wide variability could be seen amongst the sample analyzed in the present study and the other harvests, being all lower than the maximum value (5 g/100 g) required for honey freshness [40].

Generally, the nutritional value of bee pollen and honey samples can be affected by many factors such as climatic and geographic conditions, botanical origin as well as apicultural practices [59,60].

### 3.3. Mineral Content in Bee Pollen and Honey

The analysis of minerals content in bee pollen and honey is summarized in Table 6. The following minerals: potassium, calcium, magnesium, iron, and zinc were presented in bee pollen by amounts significantly higher than honey (K, 1439.80 ± 20.66 vs. 514.21 ± 3.12 mg/kg; Ca, 1011.54 ± 41.11 vs. 260.02 ± 1.23 mg/kg; Mg, 278.54 ± 13.30 vs. 52.02 ± 0.54 mg/kg; Fe, 107.16 ± 6.26 vs. 2.14 ± 0.03 mg/kg; Zn, 16.10 ± 1.77 vs. 0.75 ± 0.04 mg/kg, respectively). On the other hand, a significant concentration of sodium was detected in honey (67.82 ± 0.27 mg/kg) in comparison to bee pollen (26.99 ± 1.34 mg/kg), while no significant difference was detected in the copper content. For the toxic metals (lead and cadmium) the detected concentrations are below the recommended limits (limit values of 0.5 mg/kg for lead and 0.1 mg/kg for cadmium) [9,61]. This confirms the purity and good quality of our bee pollen and honey samples. The mineral contents in this study were within the range of the results obtained for Moroccan honey and monofloral bee pollen from different geographical origins [5,15,38].

The content of minerals in bee pollen and honey is also influenced by botanical origin, climatic conditions, and seasonal variations [62].

There is evidence that minerals are essential for the proper functioning of the body. For instance, iron plays a key role in the synthesis and function of hemoglobin [63]. Zinc and copper are essential for superoxide dismutase activity, an antioxidant enzyme used by the organism to defend against superoxide radicals [64]. Similarly, it was shown that dietary potassium intake reduces blood pressure, and plays a role in endothelial and cardiovascular function [65]. In this sense, and knowing the chemical composition of our products, we can say that they can be a good option as a food or the basis for other food products, cosmetics, or pharmaceutical formulations.

### 3.4. Phenolic Compounds in Honey and Bee Pollen

Phenolic compounds or polyphenols are a family of complex molecules widely distributed in the plant kingdom. They are categorized into phenolic acids, flavonoids, stilbenes, and lignans, among others [66].

The analysis of TPC showed that bee pollen was rich in phenolic compounds in comparison to honey (17.07 ± 0.02 mg GAE/g vs. 1.13 ± 0.00 mg GAE/g, respectively). Similarly, the TFC was significantly higher in bee pollen than in honey (4.16 ± 0.12 mg QE/g vs. 0.08 ± 0.01 mg QE/g, respectively) (Figure 1). These results were within the range of the findings in the study by Soares de Arruda [67] for Brazilian bee pollen and higher than those reported for Turkish honey [68].

Nowadays, the research of new safer, and sustainable bio-valuable molecules from functional foods is a leading tendency of green chemistry. In the present study, individual phenolic compounds in honey and bee pollen collected from the same hives were analyzed and quantified using UHPLC. The results presented in Table 7 showed that the following compounds are presented in both honey and bee pollen: o-coumaric acid, ferulic acid, ellagic acid, naringin, resveratrol, rosmarinic acid, and quercetin. These bioactive compounds were presented in bee pollen with higher amounts than honey. This is probably because the percentage of pollen grains in bee pollen is higher than in honey, and it is known that pollen grains are the main source of the phenolic compounds found in bee-hive products [69].

Some phenolic compounds are unshared between the pooled samples of honey and pollen. For instance, catechin, caffeic acid, and vanillic acid were detected only in honey in the following concentrations: 13.8 ± 0.0 mg/kg, 5.7 ± 0.0 mg/kg, and 3.6 ± 0.0 mg/kg, respectively. While, hesperidin, cinnamic acid, apigenin, rutin, chlorogenic acid, and kaempferol were detected only in bee pollen in the following concentrations: 488.9 ± 4.0 mg/kg, 150.4 ± 0.8 mg/kg, 59.2 ± 24.2 mg/kg, 182.4 ± 2.4 mg/kg, 32.1 ± 0.1 mg/kg, 4.3 ± 0.3 mg/kg, respectively. Previous studies conducted on Moroccan honey and bee pollen have reported that these two bee products are rich sources of phenolic compounds. For instance, Elamine et al. [70] have shown that *Bupleurum spinosum* honey collected from the Atlas Moroccan Mountains contains an amount of methyl syringate more than 50% of total polyphenols, and they found a correlation between this phenolic compound and the antioxidant and the antiproliferative activities. Similarly, El Ghouizi and coworkers [71] showed that Moroccan fresh bee pollen contains nineteen phenolic compounds including, ellagic acid, kaempferol glycosides, quercetin, luteolin, and isorhamnetin.

The phenolic composition of honey and bee pollen depends mainly on their floral source and also on environmental and climatic factors [72]. It has been reported that the determination of the botanical origin of honey was based on its content on some individual phenolic compounds. For instance, citrus honey can be marked by its content in hesperidin and naringin [73,74], rosemary honey is characterized by its content in 8-methoxy-kaempferol, and lavender honey is characterized by its content in luteolin [75]. This highlights the importance of individual phenolic compounds analysis as a promising tool to authenticate and predict the botanical source of organic local honey to attribute their commercial label.

A growing body of research indicates and identifies many aspects of the biological activities of phenolic compounds. For example, it was found that o-coumaric acid exhibits a potential anticancer effect via the inhibition of angiogenesis [76]. Similarly, it was shown that ferulic acid supplementation in hyperlipidemic subjects can reduce cardiovascular disease through the amelioration of oxidative stress, the improvement of lipid profiles, and inflammation [77]. The antiproliferative effect of resveratrol (stilbene) was proven via the inhibition of IGF-1R/Akt/Wnt pathways and the activation of tumor suppressor p53 protein [78]. Additionally, the anti-inflammatory and antioxidant effects were shown by apigenin, kaempferol, and quercetin (flavonol glycoside) [79]. Moreover, the intake of bee pollen and honey (rich in phenolic compounds) has shown numerous benefits for the health of consumers. For example, studies completed by several authors show that honey can prevent blood, hepatic, and renal lead toxicity [80], as well as pollen, which shows activity in the prevention/treatment of diabetes [81,82].

### 3.5. Structural Characterization by ATR-FTIR Spectroscopy

FTIR-ATR spectra of honey and bee pollen are shown in Figure 2. Both examined samples showed most of the spectral peaks in the 2000–400 cm^−1^ region and only four peaks between 4000 and 2000 cm^−1^ in which a large band at 3286 cm^−1^ corresponds to the stretching mode of OH from water [83]. As expected, this peak is more intense in honey than in bee pollen. Strong peaks between 3200 cm^−1^ and 2800 cm^−1^ were detected in honey and bee pollen but with different intensities. Indeed, bee pollen presented more intense peaks. These peaks are assigned to the CH_2_ asymmetric stretching (2919 cm^−1^) and CH_2_ symmetric stretching vibrations (2850 cm^−1^) of lipids and hydrocarbons [84]. The band at 1735 cm^−1^ is linked to the stretching mode of carbonyl moiety and asymmetric bending vibration C=O of amino acids, lipids, and flavonoids. This peak is more intense in bee pollen, which is confirmed by its high number/concentration of flavonoids as compared to the honey (see Table 7) [85]. A shoulder peak at 1635 cm^−1^ is related to the C-H deformations and aromatic stretching or frame vibration of C=O and C=C of flavonoids and asymmetrical stretching of N-H from amino acids. Absorption at 1542 and 1516 cm^−1^ is usually due to the bending mode of CH_2_ present in the chemical structure of amides II and C=C stretching vibrations of phenolic acids [83]. Moreover, peaks between 1440 and 1370 cm^−1^ represent C-H deformation vibration, OH stretching vibrations, and CH_3_ bending vibration obtained from cellulose, lipids, and functional groups ketone, aldehyde, glucose, and fructose [83]. In the fingerprint region (1200–500 cm^−1^) the intense peak with the shoulder at 1027 cm^−1^ is observed for both products, corresponding to the C–C, C–N, and C–O stretching vibrations of proteins and sugars [86]. Finally, peaks between 921 and 700 cm^−1^ resulting from vibrational modes of C-H (919 cm^−1^) and C–OH (864–767 cm^−1^) are present in the chemical structure of saccharides.

### 3.6. Antioxidant Activity

Antioxidant compounds are known to be beneficial for human health by decreasing oxidative stress and maintaining the body’s homeostasis.

The different methods used for the determination of antioxidant activity allow different mechanisms of action of natural matrices and their compounds to be evaluated. DPPH is the simplest and most widely used method for determining the free radical scavenging capacity. The ABTS assay is based on the interaction between the antioxidant and ABTS cation radical (ABTS^•+^), that, in the presence of hydrogen donating antioxidant, the ABTS^•+^ nitrogen atom quenches the hydrogen atom, causing the solution decolorization. Ferric reducing antioxidant power (FRAP assay) consists of the ability of compounds to reduce ferric ions (Fe^3+^ to Fe^2+^) in the form of ferric 2,4,6-tripyridyl-s-triazine (TPTZ), confirming the presence of reducing antioxidants [27].

Therefore, the antioxidant activity of honey and bee pollen was evaluated by three different and complementary methods (DPPH, ABTS, and FRAP assays), and the results are presented in Table 8. Concerning the DPPH test, bee pollen showed a value of IC_50_ = 50.35 ± 2.27 µg/mL, and a value of IC_50_ = 717.41 ± 7.33 µg/mL for honey. The results for the ABTS test were 397.97 ± 11.99 µg/mL and 4600 ± 70 µg/mL for bee pollen and honey, respectively. However, for the FRAP test, higher antioxidant activity (antioxidant reducing power) was also observed for bee pollen samples compared to honey samples (208.73 ± 2.04 vs. 42.74 ± 0.25 µmol Fe^2+^/g, respectively). These results are within the range of those reported for other Moroccan honey and bee pollen samples [38,71].

The obtained results revealed that the sample with the highest content of phenolic compounds is the one that has greater antioxidant activity. This goes in hand with the outcome of Adaškevičiūtė and collaborators [37] showing that bee pollen is richer in antioxidant compounds than honey, and confirmed by our results presented in Table 7 and Table 8.

### 3.7. Inhibitory Effect of Honey and Bee Pollen against α-Glucosidase and α-Amylase

Starch is a high molecular weight glucose polymer; it represents the main source of carbohydrates for humans [87]. To use starch by the organism, the digestive system breaks it down into disaccharides using α-amylase enzyme located in the brush border of the small intestine, and then into glucose using α-glucosidase enzyme present in saliva and pancreatic juice [88]. Therefore, these enzymes play an important role in the regulation of postprandial blood sugar. In some cases, the hyperactivity of these enzymes, insulin resistance, or insulin deficiency leads to hyperglycemia, and to correct this problem α-amylase or α-glucosidase inhibitors have been given orally to prevent the digestion of carbohydrates [89,90].

The results presented in Figure 3 showed that honey and bee pollen can inhibit α-amylase and α-glucosidase enzymes in vitro. The lowest IC_50_ was presented by bee pollen for α-glucosidase and α-amylase inhibitory activity (0.82 ± 0.01 mg/mL and 0.53 ± 0.01 mg/mL, respectively). While for honey the values of IC_50_ were 3.85 ± 0.12 mg/mL for α-glucosidase inhibitory activity and 2.58 ± 0.04 mg/mL for α-amylase inhibitory activity. The inhibition of these enzymes may be due to the presence of several phenolic compounds in the examined extracts. It was found that phenolic acid and flavonoid compounds displayed powerful α-amylase and α-glucosidase inhibitory activities through specific molecular interactions, more precisely by establishing hydrogen bonds between the hydroxyl groups of their aromatic ring and the active site of α-glucosidase and α-amylase [91,92].

Moreover, Tadera and coworkers [91] documented that flavonoids component belonging to flavonols and isoflavones groups exhibited more potent inhibition of both enzymes than those belonging to flavanone, flavone, and flavan-3-ol groups. This may explain the highest α-amylase and α-glucosidase inhibitory effect of bee pollen rich in flavonol molecules, especially quercetin and kaempferol as compared to their content in the honey sample.

Similarly, it was shown that the administration of resveratrol at a dose of 30 mg/kg BW in high-fat-fed mice can lower postprandial hyperglycemia [93]. Furthermore, the antihyperglycemic/antidiabetic activity of honey and bee pollen was previously confirmed to be effective in the amelioration of the rise in blood glucose levels [81,94].

## 4. Conclusions

For the first time, honey and bee pollen from the same origins were analyzed and compared, and it was confirmed that the values of studied parameters for honey and bee pollen samples respect the international regulations for these two products. These bee products show interesting nutritional capabilities due to their high carbohydrate and protein content. Moreover, from the nutritional point of view, the composition of bee pollen suggests its consideration and its incorporation as a healthier alternative protein and mineral-rich food source in the daily diet. It was also shown that both honey and bee pollen samples are rich in phenolic compounds, although belonging to different chemical groups (flavonoids, phenolic acids, flavonols, and stilbenes). These functional products have good antioxidant and anti-hyperglycemic properties that may contribute to the documented health-promoting properties of bee honey and pollen, which make these products increasingly attractive to the consumer.

The general characterization of honey and bee pollen as health-promoting foods might increase their commercial value and will have a positive impact on the basic circular economy of the rural communities where they are produced. Overall, these outcomes support the possible use of honey, bee pollen, and their antioxidant-rich extracts in the food, nutraceutical, or pharmaceutical industries as safe and sustainable sources of dietary supplements/bio-valuable components.

## Figures and Tables

**Figure 1 molecules-27-05777-f001:**
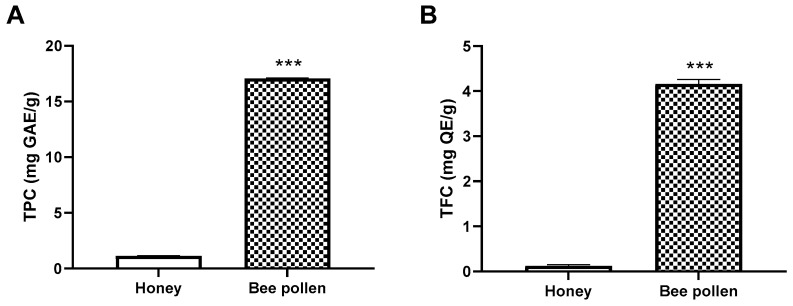
Total phenolic (**A**) and flavonoid (**B**) content of honey and bee pollen. *** *p* < 0.001 vs. Honey samples.

**Figure 2 molecules-27-05777-f002:**
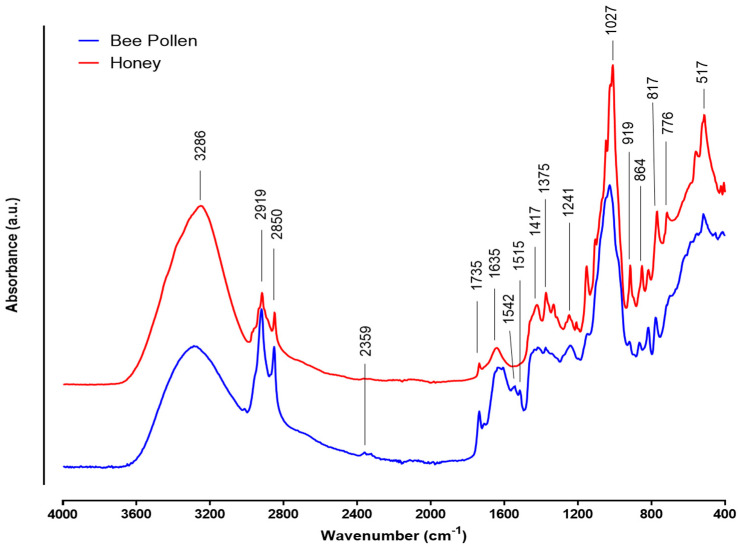
FTIR-ATR spectrum of honey and bee pollen.

**Figure 3 molecules-27-05777-f003:**
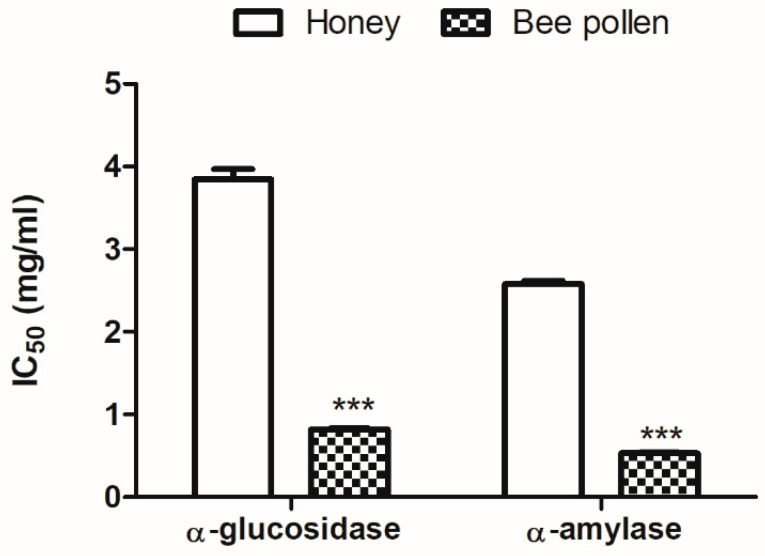
Inhibitory activity of α-glucosidase and α-amylase of honey and bee pollen. *** *p* < 0.001 vs. Honey samples.

**Table 1 molecules-27-05777-t001:** Apiary information and its geographical location.

Parameters	Information
Health status of the apiary	Free from any pathogens, disease, mite, and pesticide spray
Installation location	Boulemane (Morocco), latitude: 33°21′46.3″ N; longitude: 4°43′48.3″ W; altitude: 1752 m
Climatic conditions	Pluviometry: 9 to 60 mm; temperature: 3.2 to 22.1 °C
The bee bread used	Apis mellifera intermissa and it was placed with their queens

**Table 2 molecules-27-05777-t002:** Melissopalynological analysis of honey and bee pollen.

	Honey	Bee Pollen
**Predominant pollen (>45%)**	*Citrus aurantium* L. (Rutaceae) (48%)	*Citrus aurantium* L. (Rutaceae) (50%)
**Secondary pollen (16–45%)**	Lamiaceae (41%)	Apiaceae (33%)
**Important minor pollen (3–15%)**	Fabaceae (6%) Rosaceae (4%)	Rosaceae (15%)
**Minor pollen (<3%)**	Globulariaceae (1%)	Brassicaceae, Myrtaceae, Cistaceae, Fabaceae (2%)

**Table 3 molecules-27-05777-t003:** Physicochemical parameters of honey and bee pollen.

Parameters	Honey	Bee Pollen
Moisture (%)	20.08 ± 0.03	3.34 ± 0.02
Total soluble solids (%)	79.69 ± 0.02	-
Ash (%)	0.360 ± 0.01	3.13 ± 0.03
Total proteins (%)	0.380 ± 0.00	27.53 ± 1.71
pH	4.17 ± 0.04	4.35 ± 0.03
Electrical conductivity (µS/cm)	614.66 ± 3.78	-
Free acidity (mEq/kg)	24.13 ± 1.75	-
Lactonic acidity (mEq/kg)	9.07 ± 1.69	-
Total acidity (mEq/kg)	33.20 ± 1.76	-
Diastasic activity (Schade units/g)	12.64 ± 1.25	-
Water activity	-	0.30 ± 0.01
Pfund scale (mm)	142.78 ± 4.26	-
Melanoidins	0.93 ± 0.01	-
HMF (mg/kg)	18.63 ± 0.22	n.d.
Furfural (mg/kg)	n.d.	n.d.

n.d.: not detected; -: not analyzed.

**Table 4 molecules-27-05777-t004:** Review of the quality criteria of Moroccan honeys with a predominance of *Citrus* pollen: comparison with international standards.

Moroccan Honeys with a Predominance of Citrus Pollen									
Geographical Origin	Parameters								References
	Moisture (%)	Ash (%)	Electrical Conductivity (µS/cm)	Total Acidity (mEq/Kg)	Diastase Activity (Schade Units/g)	Carbohydrates (g/100 g)	Sucrose (g/100 g)	HMF (mg/kg)	
Berkane	14.5–21.3	-	-	12.6–44.7	1.63–29.0	-	-	5.01–43.3	[14]
North-West	14.50–21.30	-	-	12.59–44.71	1.61–287	-	0.20–5.08	5.05–43.30	[16]
Nador	15–20.19	-	192–480	11.93–50	4.3–11.00	61.73–82.55	0.23–2.52	0.08–32.60	[42]
Taza	16.71 ± 1.14	0.03 ± 0.02		16.83 ± 1.46	-	-	-	-	[12]
SidiKacem	17.2 ± 0.14	0.07 ± 0.06	87.4 ± 0.42	-	-	-	-	-	[13]
Ifrane	20.0 ± 0.1	0.15 ± 0.01	150.3 ± 2.7	30.0 ± 0.8	9.10 ± 0.47		0.97 ± 0.06	1.80 ± 1.41	[15]
**Codex Alimentarius**									
	≤20	≤0.5	≤800	≤50	≥8	≥65	≤5	≤60	[33]
**European Unit Council**									
	≤20	≤0.5	≤800	≤40	≥8	≥65	≤5	≤40	[40]

**Table 5 molecules-27-05777-t005:** Soluble proteins, total carbohydrates, and individual sugars content of honey and bee pollen.

	Honey	Bee Pollen
Soluble proteins (g BSA/100 g)	0.375 ± 0.001	27.00 ± 4.00 ***
Total carbohydrates (g GlcEq/100 g)	71.52 ± 0.33	31.69 ± 0.95 ***
Fructose (g/100 g)	36.76 ± 3.31	16.42 ± 0.09 ***
Glucose (g/100 g)	26.08 ± 2.71	11.44 ± 0.12 ***
Sucrose (g/100 g)	2.94 ± 0.35	1.38 ± 0.03 **

Values are expressed as mean ± SD. *** *p* < 0.001, ** *p* < 0.01 vs. Honey samples.

**Table 6 molecules-27-05777-t006:** Minerals content in honey and bee pollen.

Mineral Content	Concentration (mg/kg)	
	Honey	Bee Pollen
Potassium (K)	514.21 ± 3.12	1439.80 ± 20.66 ***
Calcium (Ca)	260.02 ± 1.23	1011.54 ± 41.11 ***
Sodium (Na)	67.82 ± 0.27	26.99 ± 1.34 ***
Magnesium (Mg)	52.02 ± 0.54	278.54 ± 13.30 ***
Iron (Fe)	2.14 ± 0.03	107.16 ± 6.26 ***
Copper (Cu)	0.95 ± 0.00	1.08 ± 0.05
Zinc (Zn)	0.75 ± 0.04	16.10 ± 1.77 ***
Lead (Pb)	0.03 ± 0.0	n.d.
Cadmium (Cd)	n.d.	0.013 ± 0.002

Values are expressed as mean ± SD. *** *p* < 0.001 vs. Honey samples. n.d.: not detected.

**Table 7 molecules-27-05777-t007:** Phenolic compound identification and quantification of bee pollen and honey.

Phenolic Compounds	Concentration (mg/kg)	
	Honey	Bee Pollen
Catechin	13.8 ± 0.0	n.d.
Caffeic acid	5.7 ± 0.0	n.d.
Vanillic acid	3.6 ± 0.0	n.d.
*o*-Coumaric acid	3.6 ± 0.0	39.8 ± 0.6 ***
Ferulic acid	8.3 ± 0.0	19.2 ± 0.1 ***
Ellagic acid	7.3 ± 1.3	105.1 ± 0.2 ***
Naringin	20.4 ± 0.0	49.4 ± 1.7 ***
Hesperidin	n.d.	488.9 ± 4.0
Cinnamic acid	n.d.	150.4 ± 0.8
Resveratrol	10.4 ± 0.0	157.6 ± 1.7 ***
Rosmarinic acid	15.3 ± 0.0	91.5 ± 2.2 ***
Quercetin	4.9 ± 0.0	32.1 ± 0.8 ***
Apigenin	n.d.	59.2 ± 24.2
Rutin	n.d.	182.4 ± 2.4
Chlorogenic acid	n.d.	32.1 ± 0.1
Kaempferol	n.d.	4.3 ± 0.3
**Total**	**93**	**1412**

Values are expressed as mean ± SD. *** *p* < 0.001 vs. Honey samples. n.d.: not detected.

**Table 8 molecules-27-05777-t008:** Antioxidant activity (DPPH, ABTS, FRAP) of bee pollen and honey.

	Honey	Bee Pollen
DPPH IC_50_ (µg/mL)	717.41 ± 7.33	50.35 ± 2.27 ***
ABTS IC_50_ (µg/mL)	4600.0 ± 70.2	397.97 ± 11.99 ***
FRAP (µmol Fe^2+^/g)	42.74 ± 0.25	208.73 ± 2.04 ***

Values are expressed as mean ± SD. *** *p* < 0.001 vs. Honey samples.

## Data Availability

Not applicable.

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
