# Peer review of "Exploring the Palynological, Chemical, and Bioactive Properties of Non-Studied Bee Pollen and Honey from Morocco"

_molecules, 2022, doi:10.3390/molecules27185777_

Round 1

Reviewer 1 Report

In my opinion the research was conducted correctly, with all the principles of scientific objectivity. The autors investigated the nutritional values, physicochemical characteristics, biocompounds composition (the contents of phenolics, proteins, minerals and sugar), antioxidant activity, and the in vitro inhibition of α-amylase and α-glucosidase of two bee products: honey and bee pollen.

Methodically, the work was done correctly. The description of the results and discussion are also correct, and the conclusions are adequate to the obtained results. In this context, the work

performed is complete and does not raise any objections.

From the consumer's of bee products point of view, the paper is very interesting, because it presents the rich composition of the tested honeys and pollen. Biologically active ingredients are particularly valuable, as they provide these products with nutritional, prophylactic and pro-health properties. But admittedly, that the honey and bee pollen samples were collected from the apiary in Boulemane, Morocco, which is an area rich in citrus plants, so results are regionally relevant.

In the list of references, positions no. 11, 12, and 13 (lines 570-574) are written in a different font (Calibri) than the rest of the work.

The position no. 39 in the reference list (line 625) - incorrect citation of the authors' data (first names instead of surnames).

Author Response

We would like to thank the reviewer that spent time in this process and add value to the manuscript. The manuscript has been improved according to the suggestions of reviewers; all changes were marked up using the “Track Changes” function.

     ≠Reviewer 1

Comment: In my opinion the research was conducted correctly, with all the principles of scientific objectivity. The authors investigated the nutritional values, physicochemical characteristics, biocompounds composition (the contents of phenolics, proteins, minerals and sugar), antioxidant activity, and the in vitro inhibition of α-amylase and α-glucosidase of two bee products: honey and bee pollen.

Methodically, the work was done correctly. The description of the results and discussion are also correct, and the conclusions are adequate to the obtained results. In this context, the work performed is complete and does not raise any objections.

From the consumer's of bee products point of view, the paper is very interesting, because it presents the rich composition of the tested honeys and pollen. Biologically active ingredients are particularly valuable, as they provide these products with nutritional, prophylactic and pro-health properties. But admittedly, that the honey and bee pollen samples were collected from the apiary in Boulemane, Morocco, which is an area rich in citrus plants, so results are regionally relevant.

Response: Thank you for your positive opinion.

Comment: In the list of references, positions no. 11, 12, and 13 (lines 570-574) are written in a different font (Calibri) than the rest of the work.

Response: The entire list of references has been rewritten according to the recommended font type.

Comment: The position no. 39 in the reference list (line 625) - incorrect citation of the authors' data (first names instead of surnames).

Response: The reference was corrected.

Reviewer 2 Report

In the manuscript submitted the authors have conducted a study with the aim of presenting the physicochemical composition, nutritional value and phenolic profile of a pooled sample of bee pollen and honey from Boulemane, Morocco. This article about Exploring the palynological, chemical and bioactive properties of non-studied bee pollen and honey from Morocco by Bakour et al. contains a great number of analyses have been done. Unfortunately, the submitted manuscript has significant flaws, so in my opinion it should not be accepted for publication.

In the introduction part, the authors mention only one paper related to the analysis of pollen from the geographical area of Morocco. It is necessary for the authors to give a review of the analyzes of honey and pollen, and concerning citrus, which were carried out in various earlier studies from the area of Morocco.

In the part of the methodology that refers to the formation of the initial pooled sample, it is necessary to describe precisely how a unique sample of honey, i.e. pollen, was formed. In my view the authors must have stated the time and conditions of storing of samples from 2019 to the moment of analysis which could have affected the results obtained.

Pollen pH was measured using a water-ethanol solution. It is necessary to explain how the pH-value was corrected in relation to the water system.

In the part of the methodology related to the identification and quantification of individual phenolic components, it is necessary to specify the standards based on which the identification of phenolic compounds was carried out.

The applied statistical method using the Student's t-test in the comparison of parameters within one pooled sample of honey and pollen from one apiary cannot be a basis for making conclusions about the significance of the statistical difference in the examined physico-chemical and other parameters.

The results of the physicochemical analysis of pooled samples of honey and pollen, obtained in the manuscript, suggest a standard laboratory report on the analysis performed in accordance with the current legislation. In the case of a pooled sample of honey, the water content even slightly exceeds the legal limit of twenty percent. The presented physical and chemical parameters do not provide new findings.

Conclusion has no clearly highlighted novelties.

L48 Replace the word "makers" with the word "markers".

L122 to L124 It is recommended not to begin the sentence with a numerical value. The paragraph should be restyled.

L123 It needs to accurately specify the equipment and manufacturer of used apparatus to determine the color of the honey.

L128 It is necessary to state accurately the equipment and manufacturer of used apparatus to determine the melanoidins content of honey.

L143 It is necessary to state accurately the equipment and manufacturer of used apparatus to determine the mineral content.

L281 In Table 2, the taxonomic classification of the melissopalynological analysis of honey and pollen is shown with the dominant pollen of Citrus aurantium L. at the species level as the predominant pollen. The presence of pollen was then classified into secondary, important minor and minor pollen at the family level. I suggest that, except for the predominant pollen and for other categories of pollen, the taxonomic classification, where possible, should be presented at the subspecies level, i.e. in more detail.

L296 I suggest that the term “go in hand” be replaced by the term “are in agreement”.

L351 Replace value 2.93 ± 0.35 g/100g with value 2.94 ± 0.35 g/100g  from the table 4.

L353 Replace value 36.76 ± 3.30 g/100g with value 36.76 ± 3.31 g/100g  from the table 4.

L407 to L409 The sentence from 407 to 409 is not clear enough, please rephrase it.

L442 In Table 2 replace the term “ferrulic acid” with the term “ferulic acid”.

L570-L574 Match the type of font to the recommended type of font.

L609 The reference should be marked with serial number 31.

L616 The reference under number 34 is not mentioned in the text.

Author Response

We would like to thank the reviewer that spent time in this process and add value to the manuscript. The manuscript has been improved according to the suggestions of reviewers; all changes were marked up using the “Track Changes” function.

     ≠Reviewer 2 

Comment: In the introduction part, the authors mention only one paper related to the analysis of pollen from the geographical area of Morocco. It is necessary for the authors to give a review of the analyzes of honey and pollen, and concerning citrus, which were carried out in various earlier studies from the area of Morocco.

Response: Thank you for your comment, more detail was added in the introduction part concerning the honey bee pollen and citrus from the area of Morocco. Please see from line 54 to line 62, page 2.

In the introduction part, we mention only one paper related to the analysis of pollen from the geographical area of Morocco because this is the only published work investigating the nutritional quality and the physicochemical characterization of 8 monofloral bee pollens collected from different localities of Morocco.

Concerning the review of the analyzes of honey, pollen, and citrus, we added a Table to review the quality criteria of Moroccan honey with a predominance of citrus pollen in comparison with international standards. Please see Review Table 4, page 10.

Comment: In the part of the methodology that refers to the formation of the initial pooled sample, it is necessary to describe precisely how a unique sample of honey, i.e. pollen, was formed. In my view the authors must have stated the time and conditions of storing of samples from 2019 to the moment of analysis which could have affected the results obtained.

Response: The time and conditions of storing samples from 2019 to the moment of analysis were added in lines 75 to line 77, page 2.

Comment: Pollen pH was measured using a water-ethanol solution. It is necessary to explain how the pH-value was corrected in relation to the water system.

Response: The pH was conducted on ultrapure water solution, the sentence “or in hydroethanolic bee pollen solution 10% (w/v)”, was a question of a co-author in the initial draft and by mistake was not deleted from the final manuscript. For that, we dropped it from the current version, please see lines 98-101, page 3. 

Comment: In the part of the methodology related to the identification and quantification of individual phenolic components, it is necessary to specify the standards based on which the identification of phenolic compounds was carried out.

Response: The standards based on which the identification of phenolic compounds was carried out were added from line 205 to line 210, page 5.

Comment: The applied statistical method using the Student's t-test in the comparison of parameters within one pooled sample of honey and pollen from one apiary cannot be a basis for making conclusions about the significance of the statistical difference in the examined physico-chemical and other parameters.

Response: The Student's t-test is one of the most commonly used statistical tests when it comes to comparing the means of two small samples. And it is used to make the significance of the statistical difference

Please see the following reference

Reference 1: Marino, M. J. (2018). Statistical analysis in preclinical biomedical research. In Research in the biomedical sciences (pp. 107-144). Academic Press.

Reference 2: Feng, Y. C., Huang, Y. C., & Ma, X. M. (2017). The application of Student’s t-test in internal quality control of clinical laboratory. Frontiers in Laboratory Medicine, 1(3), 125-128.

Reference 3: Nayak, B. K., & Hazra, A. (2011). How to choose the right statistical test?. Indian journal of ophthalmology, 59(2), 85.

Comment: The results of the physicochemical analysis of pooled samples of honey and pollen, obtained in the manuscript, suggest a standard laboratory report on the analysis performed in accordance with the current legislation. In the case of a pooled sample of honey, the water content even slightly exceeds the legal limit of twenty percent. The presented physical and chemical parameters do not provide new findings.

Response: The discussion part of the results of the physicochemical analysis of pooled samples of honey and pollen was improved to highlight the importance of these analyses for the labeling and commercialization of these functional foods.

Concerning the water content in honey, the sentence was corrected.

Comment: Conclusion has no clearly highlighted novelties.

Response: The conclusion part was improved. Please see from line 646 to line 663, page 17.

Comment: L48 Replace the word "makers" with the word "markers".

Response: Done, please see line 50, page 2.

Comment: L122 to L124 It is recommended not to begin the sentence with a numerical value. The paragraph should be restyled.

Response: The paragraph was restyled.

Comment: L123 It needs to accurately specify the equipment and manufacturer of used apparatus to determine the color of the honey.

Response: The equipment and manufacturer of the used apparatus to determine the color of the honey was added. Please see lines 136-139, page 3.

Comment: L128 It is necessary to state accurately the equipment and manufacturer of used apparatus to determine the melanoidins content of honey.

Response: The equipment and manufacturer of used apparatus to determine the melanoidins content of honey was added. Please see lines 144-146, page 4.

Comment: L143 It is necessary to state accurately the equipment and manufacturer of used apparatus to determine the mineral content.

Response: The equipment and manufacturer of the used apparatus to determine the mineral content was added to the methodology. Please see lines 160-162, page 4.

Comment: L281 In Table 2, the taxonomic classification of the melissopalynological analysis of honey and pollen is shown with the dominant pollen of Citrus aurantium L. at the species level as the predominant pollen. The presence of pollen was then classified into secondary, important minor and minor pollen at the family level. I suggest that, except for the predominant pollen and for other categories of pollen, the taxonomic classification, where possible, should be presented at the subspecies level, i.e. in more detail.

Response: According to the standard reference used for the melissopalynological analysis ‘ If detailed knowledge is not available, it is possible to associate the pollens in larger groups” as presented in the manuscript for secondary pollen, important minor pollen, and minor pollen.

Comment: L296 I suggest that the term “go in hand” be replaced by the term “are in agreement”.

Response: Thank you for your suggestion.

Comment: L351 Replace value 2.93 ± 0.35 g/100g with value 2.94 ± 0.35 g/100g from the table 4.

L353 Replace value 36.76 ± 3.30 g/100g with value 36.76 ± 3.31 g/100g from the table 4.

Response: The values have been replaced in the manuscript. Please see Table 5, page 11.

Comment: L407 to L409 The sentence from 407 to 409 is not clear enough, please rephrase it.

Response: The sentence has been changed.

Comment: L442 In Table 2 replace the term “ferrulic acid” with the term “ferulic acid”.

Response: This has been corrected. Please see Table 7, page 13.

Comment: L570-L574 Match the type of font to the recommended type of font.

Response: The entire text has been rewritten according to the recommended font type.

Comment: L609 The reference should be marked with serial number 31.

Response: The entire list of references has been revised.

Comment: L616 The reference under number 34 is not mentioned in the text.

Response: This reference was removed from the list of references.

Round 2

Reviewer 2 Report

In the submitted revised manuscript, the authors conducted research to present the physicochemical composition, nutritional value, and phenolic profile of a bulk sample of bee pollen and honey from Boulemane, Morocco. This article on the investigation of palynological, chemical and bioactive properties of unexplored bee pollen and honey from Morocco, Bakour et al. contains a large number of performed analyses. After corrections have been made based on the revision, I am of the opinion that the manuscript can be published.